# Study of Meat and Carcass Quality-Related Traits in Turkey Populations through Discriminant Canonical Analysis

**DOI:** 10.3390/foods12203828

**Published:** 2023-10-19

**Authors:** José Ignacio Salgado Pardo, Francisco Javier Navas González, Antonio González Ariza, José Manuel León Jurado, Inés Galán Luque, Juan Vicente Delgado Bermejo, María Esperanza Camacho Vallejo

**Affiliations:** 1Department of Genetics, Faculty of Veterinary Sciences, University of Córdoba, 14071 Córdoba, Spain; josalgadopardo@outlook.com (J.I.S.P.); fjng87@hotmail.com (F.J.N.G.); inesgalanluque@gmail.com (I.G.L.); juanviagr218@gmail.com (J.V.D.B.); 2Agropecuary Provincial Centre, Diputación Provincial de Córdoba, 14071 Córdoba, Spain; jomalejur@yahoo.es; 3Institute of Agricultural Research and Training (IFAPA), Alameda del Obispo, 14004 Córdoba, Spain; mariae.camacho@juntadeandalucia.es

**Keywords:** *Meleagris gallopavo*, product characterization, breed differences, carcass yield, meat nutrients

## Abstract

The present research aimed to determine the main differences in meat and carcass quality traits among turkey genotypes worldwide and describe the clustering patterns through the use of a discriminant canonical analysis (DCA). To achieve this goal, a comprehensive meta-analysis of 75 documents discussing carcass and meat characteristics in the turkey species was performed. Meat and carcass attributes of nine different turkey populations were collected and grouped in terms of the following clusters: carcass dressing traits, muscle fiber properties, pH, color-related traits, water-retaining characteristics, texture-related traits, and meat chemical composition. The Bayesian ANOVA analysis reported that the majority of variables statistically differed (*p* < 0.05), and the multicollinearity analysis revealed the absence of redundancy problems among variables (VIF < 5). The DCA reported that cold carcass weight, slaughter weight, sex-male, carcass/piece weight, and the protein and fat composition of meat were the traits explaining variability among different turkey genotypes (Wilks’ lambda: 0.488, 0.590, 0.905, 0.906, 0.937, and 0.944, respectively). The combination of traits in the first three dimensions explained 94.93% variability among groups. Mahalanobis distances cladogram-grouped populations following a cluster pattern and suggest its applicability as indicative of a turkey genotype’s traceability.

## 1. Introduction

Poultry has an advantaged position over other livestock in the worldwide meat industry [1]. This is due to its healthy characteristics and affordability, as well as the lack of religious restrictions [2,3]. The poultry meat industry has exponentially developed over the past 40 years and is expected to keep growing until 2050, especially in developing countries [1]. For this reason, while global meat consumption decreased by 0.601 kg per capita during the 2017–2022 period, poultry meat consumption increased by 0.722 kg per capita [4], as shown in Figure 1.

The worldwide turkey industry has historically been focused on achieving the greatest meat productivity at the lowest costs [1,5]. This need has led to intense selection for fast growth and high-value piece yield and, therefore, heavy and ‘double breast’ commercial strains have been developed [3]. This fact was evidenced by the doubling of the weight of turkey poults at the same age between 1966 and 2003 [6]. However, the exhausting selection for growth and the intensification of production systems lead to the apparition of carcass incidences and meat quality aberrations [7,8], and concerns about meat quality emerged among consumers [7]. Costumers’ perceptions of meat quality reside in the appearance of the product at the point of purchase and the texture of the cooked meat [3]. However, poultry meat quality research also includes physical, chemical, packaging, and health-related traits [3]. In fact, quality studies have developed from simple measures (such as carcass weight or meat pH) to highly sophisticated measures, such as muscle fiber diameter [9] or the amino acid profiles of specific carcass cuts [10]. Nevertheless, the current trend in meat quality research only analyzes meat texture, nutrient composition, and muscle color [11]. This could be due to the fact that most common meat defects in poultry negatively affect meat color and water-holding capacity, which have been related to those traits crucial for consumer acceptance [12].

Measures adopted to prevent carcass defects and improve meat quality have been focused on housing, management, and slaughter practices [3]. However, as these measures are a consequence of genetic targeting for growth, meat, and carcass quality, they could be improved through selection [13]. Meat color, pH, and water-holding capacity are traits that have shown correlation with certain myopathies [3]. Hence, selection for these traits might improve meat quality and reduce the incidence of myopathies such as pale, soft, and exudative (PSE) syndrome. The employment of this quality-related targeting might also be beneficial in local breeds, as it could improve the efficiency of conservation and breeding programs based on the profitable sustainability and quality of their products [11].

Nevertheless, the realistic purpose of a turkey population should be considered before developing any kind of meat quality selection program [3]. In the commercial turkey industry, two lines are commonly distinguished: a ‘sire’ line focused on fast growth and a ‘dam’ line focused on reproductive traits [3,14]. However, those heavy strains are selected for the yield of valuable parts at quartering and meat processing [15,16,17]. The growing market of organic, slow-growing systems is occupied by small hybrid strains and light worldwide-spread breeds [18]. They usually dominate the whole-bird market through their massive sales during festivities such as Christmas or Thanksgiving [17]. On the other hand, native poultry breeds can be found in the backyard and extensive systems as dual-purpose (meat and eggs) genotypes [19]. They are locally adapted and highly rustic [20], maintaining ancestral behaviors [21] and agility for flying and roosting in trees to avoid predators [11]. Their production is mainly for self-consumption associated with local festivities [20], elaborated by following traditional cuisine recipes [22].

Although the study of meat and carcass quality has been focused on economic and productive goals, there are a few comparison studies among turkey populations. Some studies have described differences not only among breeds but also among lines of commercial strains [3,14,23,24]. However, other studies showed no or limited differences when comparing industrial hybrids [6,17]. There are no further studies on this subject, and when considering local genotypes, the absence of comparative studies is even more remarkable. This lack of attention to native breeds has also been evidenced in the lack of productive characterization studies which, moreover, are crucial for their official recognition and the implementation of conservancy proceedings [20]. While some meta-analyses have been carried out on meat and carcass quality traits in chicken local genotypes [11,25], no research has been conducted on the turkey species.

Thus, the present study aims to identify the differences in carcass and meat quality traits among worldwide turkey breeds and commercial strains, as well as whether they respond to a clustering pattern. Discriminant canonical analysis (DCA) was used to design a statistical tool that allows us to determine if a specific carcass or meat piece fit the features of the different breeds included. Results obtained from the present work would improve the efficiency of conservancy actions and breeding programs of native turkey breeds and could help in the design of future research focused on turkey meat and carcass quality studies.

## 2. Materials and Methods

### 2.1. Systematic Review Approach Decision

The approach to this study has been previously described as an efficient tool in the animal science field for specific subjects [11,25,26,27]. Due to the healthcare focus of the PRISMA guidelines for systematic reviews, this method does not perform properly for the wide range of papers publishing livestock and local breeds’ information [28]. Moreover, a faithful adscription to PRISMA’s guidelines did not reported changes in the level of recommendation and endorsement of the journal [29] and showed limited applicability in conservancy and environmental management reviews [30].

### 2.2. Data Collection

Data collection was performed following the guidelines described by previous authors [11,26,27]. For this purpose, the repositories at www.google.scholar.es and www.sciencedirect.com (visited on 31 November 2021) were employed. On the other hand, other information sources as www.ncbi.nlm.gov/pubmed/ (accessed on 20 August 2023) were discarded since they do not enable data extraction for analysis [11,27]. Therefore, the public filters that can be implemented were not comprehensive enough to perform the adequate analysis required for this retrospective observational longitudinal study over the period extending from 1968 to 2021, inclusive of both years. ‘Meat quality’, ‘carcass quality’, and ‘carcass traits’ were used as keywords in the research, with each one followed by ‘turkey’, ‘Meleagris gallopavo’, or any semantically related term [31]. A total of 75 documents were found and included in this study. As mentioned above, all papers collected were published from 1968 to 2021. The fact that some old papers have been used in the present work is proof that the number of studies was not very abundant (75 papers), and the inclusion of all of them enriches the information available on the different turkey genotypes. All the publications included in the present study were written in the English language. Table 1 shows all parameters of carcass and meat traits included in each cluster for the analysis and the references of the studies used in each cluster.

A total of 889 observations were individually recorded, considering the meat cut from which they were obtained and the number of samples. Meat cuts and carcass components from which observations were sampled were carcass reminder, breast, complete leg, thigh, drumstick, wings, head, neck, feet, shank, back, heart, liver, giblets, kidney, lungs, spleen, pancreas, gallbladder, proventriculus, gizzard (full and empty), stomach, complete intestine, small intestine, cecum, abdominal fat, fat pad, ovary, oviduct, feathers, skin, feather plus skin, blood, and waste. After performing the document evaluation, a total of 22 dependent variables were considered in the statistical analysis: carcass/piece weight, carcass/piece yield, cold carcass weight, slaughter weight, muscle fiber diameter, pH, pH 24 h, L* meat, a* meat, b* meat, drip loss, water-holding capacity, cooking loss, shear force, springiness, fragmentation index, moisture, protein, fat, ash, collagen, and cholesterol. The methodologies used for the determination of each particular explanatory variable in each specific document were not registered, as the techniques and procedures followed during measurement collection were standardized to be considered in the research procedures. Additionally, this decision was made on the basis that when standardized techniques are used, even if differences across methods and procedures may exist, these are negligible, as supported by scientific evidence. 

Possible differences in units used to quantify carcass and meat quality across the literature were neutralized by applying the corresponding conversion to make observations across papers comparable. All units were converted to the units most frequently used across the documents (Table 1).

A total of 9 turkey breeds were distinguished in the papers, including Beltsville Small White, Egyptian local turkey, Nigerian local turkey, Lebanese local turkey, Turkish Bronze turkey, North Caucasian Bronze turkey, ‘Wild turkey’, Commercial, and Commercial (Unspecified). ‘Aviagen’ and ‘Nicholas-BUT’, which were also included as breeds in the bibliography, were removed from the analysis due to them each only contributing one observation. 

### 2.3. Data Analysis

#### 2.3.1. Normality and Bayesian ANOVA Tests

The Shapiro–Francia W’ test was used to discard gross violations of the normality assumption because the number of observations included in the analysis was more than 50 and less than 2500. The Shapiro–Francia W′ test was performed using the Shapiro–Francia normality routine of the test and distribution graphics package of the Stata Version 16.0 software (College Station, TX, USA). All variables were non-normally distributed, except for pH, pH24, b* meat, and cooking loss, which were normally distributed (*p* > 0.05). Therefore, a Bayesian ANOVA was used on all variables (normal and non-normal) to detect differences in the median across genotypes and reported medians that significantly differ in terms of the majority of possibilities. 

The Bayes factor (BF) is able to quantify the strength of the evidence for the null and alternative hypotheses and can be used by researchers to issue conclusions, instead of frequentist *p* values. When the BF increases, the degree to which the evidence favors the alternative hypothesis compared to the null hypothesis also increases. Previous authors have proposed to favor the interpretability of the results, a method used to extrapolate between the BF used in Bayesian approaches and the *p*-values of frequentist approaches [103]. The posterior descriptive sample statistics are modeled from the means and variances of the measured unpaired groups and provided as sources of variation, while the prior element was modeled as a non-informative prior using the Jeffreys–Zellener–Siow method or, equivalently, from the calculation of a reference prior based on a gamma distribution with a standard error of 1. The 95% credibility shows that there exists a 95% probability that these regression coefficients (the posterior distribution mean value for each covariate and factor) in the population lie within the corresponding credibility intervals [104,105]. As 0 is not contained in the credibility interval, a significant effect is detected for that factor. The integral calculation of the factors is necessary for the accuracy of the BF. Subsequently, it can be used as an accurate statistical index to measure the amount of support for H_1_ (the difference between unpaired means is greater than zero) or H_0_ (the difference between unpaired means is not greater than zero). Contextually, Bayesian approaches provide better insights into the model structures of H_1_ and H_0_. Hence, the highest probabilities within likelihood distributions may not necessarily align with the average outcome of conventional tests. This observation is particularly relevant in the context of biological inference, as biological likelihoods are better equipped to address biological inquiries than numerical averages derived from unrepresentative subsets.

Those variables not obtaining significant differences were pH (F = 0.312; Pv = 0.734), drip loss (F = 0.082; Pv = 0.777), water-holding capacity (F = 1.464, Pv = 0.241), cooking loss (F = 1.965; Pv = 0.132), shear force (F = 0.446, Pv = 0.644), moisture (F = 0.310; Pv = 0.905), ash (F:1.006; Pv = 0.397), and collagen (F = 35.764; Pv = 0.105). For these variables, Bayesian inference for ANOVA was performed using the Bayesian Package of SPSS version 26.0 software (IBM, Armonk, NY, USA). 

Linear discriminant function analyses are well-known methods for distinguishing between classes of observations and constructing discriminant functions that are used for the classification of observations of unknown class membership. In addition to this approach, DCA was used to determine the minimum number of dimensions needed to describe these differences [106,107]. Hence, in the present study, the presence of differences in some variables across breeds justified the employment of a DCA. 

#### 2.3.2. Multicollinearity Preliminary Testing

In order to ensure independence and discard strong linear relationships across predictors, a multicollinearity analysis was run before statistical analyses, according to González Ariza et al. [11]. This analysis aimed to detect noise or redundancy problems in the variables used before data manipulation and exclude unnecessary variables. In this sense, multicollinearity analysis is used to avoid the over-inflation of the variance explanatory potential due to the inclusion in the analysis of an unnecessarily large number of variables [108]. The variance inflation factor (VIF) was used as a multicollinearity indicator and calculated via the use of the following formula: $$\mathrm{VIF}\;=1/\left(1-{\;R}^{2}\right),$$
where R^2^ is the coefficient of determination of the regression equation. 

The literature recommends considering a maximum VIF value of 5 [109] and a minimum tolerance value of 0.20 [110]. The multicollinearity statistics routine of the describing data package of SPSS version 26.0 software (IBM, Armonk, NY, USA) was used to perform the multicollinearity test.

#### 2.3.3. DCA

To perform the DCA, the breed used in each study found in the literature was used as the dependent variable. The aforementioned 22 parameters describing meat and carcass traits were used as explanatory variables. The sex of the individual whose each carcass piece analysis was being performed was used as the labeling classification criteria to measure the variability in quality-related traits between and within classification groups to establish, identify, and outline clusters [111]. 

The statistical analysis issued a set of discriminant functions that could be used as a tool to determine the clustering patterns described by the sample through a linear combination of carcass- and meat quality-related traits. Regularized forward stepwise multinomial logistic regression algorithms were used to perform the variable selection, following indications of González Ariza et al. [112]. 

In the present work, the choice of performing a stepwise forward analysis was made after considering the following alternatives: 

The first option considered was to perform a regularized DCA. The regularization improves the estimation of covariance matrices in situations in which the number of predictors is larger than the number of data, since, in such cases, regularization can lead to an improvement in the efficiency of the discriminant analysis. However, this was not true in our case, since the nature of the variables considered can lead to the appearance of considerable multicollinearity problems.

Such multicollinearity problems may derive from the fact that some of the variables initially considered were computed by including others (which were also included) among the terms of their formulas. As a result, even if the models were simplified, the eliminated variables may still be considered in some way.

Priors were regularized following the group sizes computed using the prior probability option in SPSS version 26.0 software (IBM, Armonk, NY, USA) instead of considering them to be equal, thus preventing groups with different sample sizes from affecting the quality of the classification [113].

#### 2.3.4. DCA Efficiency and Analysis Model Reliability

Variables that contributed to the discriminant function were evaluated via Wilks’ lambda test [111]. Despite ideal Wilks’ lambda values tending toward 0, discriminant functions under 0.05 were usually accepted [114].

Pillai’s trace criterion was the only acceptable test able to evaluate the assumption of equal covariance matrices in cases of unequal sample sizes [115]. This test was performed using the Multivariate routine of the General Linear Model package of the SPSS version 26.0 software, and statistical differences under 0.05 in the dependent variables across the levels of independent variables were accepted [111].

#### 2.3.5. Variable Dimensionality Reduction

The overall variables were minimized to a few significant variables that contributed most to the different variations in the different types of carcasses via a preliminary principal component analysis (PCA), according to the bibliography in [111]. Through this multivariate statistical approach, the variance in the sample was partitioned into a between-group and within-group component to maximize discrimination between groups. In the present study, data were first transformed using a PCA, and, subsequently, clusters were identified using a DCA.

Therefore, PCA was primarily used for dimensionality reduction and feature extraction. This method is not typically employed to directly identify clusters or groups in the data. Instead, PCA transforms the original data into a new set of variables (principal components) that are linear combinations of the original variables. These components capture most of the variance in the data, which can be useful for visualization or as inputs to other clustering methods. PCA does not inherently identify clusters but reduces the dimensionality of the data.

On the other hand, DCA is a supervised technique used when you have pre-defined groups or labels. This technique seeks to find linear combinations of variables that maximize the separation between these pre-defined groups. Unlike PCA, which is unsupervised, DCA is used to identify clusters or groupings based on the known group labels. The justification for using PCA followed by DCA is supported by three main points: the dimensionality reduction, the data transformation, and the data preprocessing. If the original data have a high number of variables, PCA can be used to extract the most informative components. This makes the subsequent application of DCA more computationally efficient and helps in visualizing the data. Moreover, PCA can transform the data in a way that highlights the dominant sources of variation, which is useful before applying DCA, especially if the data are high dimensional. Lastly, the use of PCA is part of a larger data preprocessing pipeline. For instance, this analysis can be used to deal with multicollinearity, reduce noise, or improve the data’s suitability for DCA.

#### 2.3.6. Canonical Coefficients and Loading Interpretation and Spatial Representation

The assignment percentage of the carcass or carcass piece quality within its group (defined by breed) was computed through the use of a discriminant function analysis. The variables that showed an absolute value of a discriminant loading higher than 0.40 were considered to be considerably discriminant, according to González Ariza et al. [111]. In this line, the discriminant ability was evaluated by attending to the absolute coefficients of each particular variable within a set [116]. Consecutively, squared Mahalanobis distances were computed via the use of the following formula:(1)$$D_{\mathrm{ij}}^{2}=\left(\bar{Y_{i}}-\bar{Y_{j}}\right)\;{\mathrm{COV}}^{-1}\left(\bar{Y_{i}}-\bar{Y_{j}}\right)$$
where D^2^_ij_: distance between populations i and j; Ȳ_i_ and Ȳ_j_: means of variable x in the ith and jth populations, respectively; COV^−1^: inverse of the covariance matrix of the measured variable x [114].

The squared Mahalanobis distances were graphically represented in clustering patterns defined by the differences in the values for meat and carcass quality traits across the potential classification. Thus, a dendrogram depicting the different categories within carcass and meat quality classifications was designed through the use of the underweighted pair-group method arithmetic averages (UPGMA) from the Universität Rovira i Virgili (URV), Tarragona, Spain, and the Phylogeny procedure of MEGA X 10.0.5 (Institute of Molecular Evolutionary Genetics, The Pennsylvania State University, State College, PA, USA).

#### 2.3.7. Discriminant Function Cross-Validation

To validate the discriminant functions used, the leave-one-out cross-validation approach was employed, with targeting and accuracy being at least 25% higher than those of values obtained by chance [111].

The discriminating power of the cross-validation function was compared through the use of Press’ Q significance test following the below formula:$${\mathrm{Press}}^{’}Q={\left[N-\left(\mathrm{nK}\right)\right]}^{2}\;/\;\left[N\left(K-1\right)\right]$$
where N is the number of observations in the sample; n is the number of observations correctly classified; and K is the number of groups. Subsequently, the value of Press’ Q statistic was compared to the critical value of 6.63 for χ^2^ with one degree of freedom in a significance of 0.01, meaning that a Press’ Q exceeding 6.63 classifications was considered significantly better than values obtained by chance [111].

## 3. Results

### 3.1. DCA Model Reliability

Pillai’s trace criterion described a significant difference between the different meat and carcass attributes’ classification groups (*p* < 0.05; Table 2).

The preliminary test reported no multicollinearity problems due to the fact that all variables showed VIF values under five. Hence, all quality attributes were included in further analysis. Values of tolerance and VIF for each variable are shown in Table 3. 

### 3.2. Canonical Coefficients, Loading Interpretation, and Spatial Representation

Eight discriminating canonical functions made up the DCA (Table 4). Functions F1, F2, and F3 contributed 94.93% to the explanation of the whole variance, while the remaining variables have a low percentage of explanatory ability.

Thus, the loading values of each dependent variable in the three first functions (F1, F2, and F3) are shown in Figure 2:

The test of the equality of the groups’ means across carcass and meat quality traits was used to categorize variables based on their discriminating capacity. Moreover, through the test of equality of group means, a series of variables were selected, as they obtained significant values (*p* < 0.05; Table 5). Variables were ordered in descending order for Wilks’ lambda value and in ascending order for values of F. This is due to the fact that greater values of F and lower values of Wilk’s lambda are indicators of a better discriminating ability [111]. The relative weight of each meat piece or carcass trait across the discriminant functions was measured using the standardized discriminant coefficients (Figure 3).

The substitution of the values obtained for meat- and carcass quality-related traits into the first two discriminating functions was performed to obtain the x- and y-axis coordinates (F1 and F2 functions, respectively). Thus, centroids corresponding to each breed group were depicted on a territorial map (Figure 4).

Mahalanobis distances were used due to their ability to represent the probability of matching an unknown observation to a particular classification group—in this case, they were related to the carcass and meat characteristics. In this way, the relative distance of the problem meat piece or carcass to the mean of its closest group was considered. Hence, the likelihood of matching an observation into a group was estimated, as described by Hair et al. [117]. Mahalanobis distances obtained were graphically represented on a dendrogram (Figure 5).

### 3.3. Discriminant Function Cross-Validation

Classification and leave-one-out cross-validation matrices were assessed and showed an average of 81.62% correctly assigned observations. However, great differences in the classification efficiency were observed across breeds, obtaining a 100% rate of accuracy in the Nigerian local turkey breed, while this tool had 0% efficiency in its observations of the Lebanese local turkey breed (Figure 6).

## 4. Discussion

Attending to the traits included in this study, variables not reporting medians to significantly differ were pH, water-captivity traits (drip loss, water-holding capacity, and cooking loss), shear force, and some chemical composition attributes (moisture, ash, and collagen). Meat pH, water holding capacity, and shear force are closely related traits [14]. Rapid pH decline results in a reduced water-holding capacity, which can hence result in greater meat toughness [118], drip loss, and cooking loss [14]. However, even though these traits are correlated, they did not all exhibit differences in studies comparing turkey breeds [3,6,14,17,23,24]. Concerning meat chemical composition, variance across turkey genotypes has also shown conflicting results [6,17,23,34,119]. Therefore, no clear conclusion can be drawn. Despite the differences in meat quality traits that have been reported in the chicken species, there is a lack of consensus on the turkey species [14]. Thus, further research on this matter is needed [3]. These differences in results obtained across different studies have also been widely considered. Previous authors stated that as wing flapping during slaughter influences pH and its related traits, a serial slaughter design should be developed in each breed considered in comparative studies to avoid the influence of the different handling traits [14,120]. These authors also pointed out the different availability of meat samples across breeds as a possible reason for inconsistent results [14,120]. Moreover, Werner et al. [17] reported that the variability in the slaughter age and husbandry conditions and the different post mortem times at which measurements were taken can influence the meat quality traits. In this respect, a scientific consensus should be driven to avoid external effects on the study of genotypes’ influence on meat and carcass quality.

In the present study, cold carcass weight showed the best discriminating abilities. This measurement is a commonly included factor in the different studies analyzed in the present work. Cold carcass weight can monitor evaporative losses during carcass refrigeration [121] and is widely used for estimating a carcass’ dressing percentage [122]. In poultry species, cold carcass weight is highly influenced by age, sex of the animal, rearing system, and cooling process, as well as the genotype [122,123,124]. Differences across breeds have been reported in geese [122], Japanese quail [123], and fowl [125]. These studies reported that the variance caused by genotype may be due to different muscle activity during cooling and unequal slaughter weights. 

Slaughter weight was the second highest ranked variable according to its discriminating abilities. This management factor plays a pivotal role for poultry producers from an economic point of view [126] and has been the main objective of the selection of different genotypes [127]. Thus, live weight is commonly used as a classification criterion in the different commercial turkey strains [128], as well as as an indicator of other related carcass traits [127,129].

In this way, the results obtained in this study suggest that cold carcass weight showed a greater discriminant capacity than slaughter weight. Portillo-Salgado et al. [130] claimed that slaughter weight explained by itself 95% of the variation in cold canal weight. However, differences across breeds could arise when facing the post-slaughter cooling process. Carcass weight changes during chilling are due to water content losses [131]. This range of variation could be conditioned by breed differences in the protein and moisture composition of the meat [132], the intensity of post mortem muscle contraction and pH decline [133], or the thickness of subcutaneous fat cover [134].

The variable sex-male also had good discriminatory power. This finding might suggest the profitability of recording the sex of the individual in meat and carcass studies due to potential sexual dimorphism across breeds. In this respect, a greater sexual dimorphism in body weight has been described in the Tunisian [135], Nigerian [136], and Andalusian [137] turkey breeds compared to those reported in commercial strains [138,139]. This would suggest a greater sexual dimorphism in carcass characteristics in primitive or unselected populations compared to those commercial strains. 

Carcass or piece weight was also a variable that showed big differences across breeds. Carcass components grow in parallel with live weight as the animal ages [127,129]. However, selection for greater development of certain body parts [130] and in premature stages of life [140] has been performed. This is due to the great value of and demand for primary cuts in the market [129]. Different weights of carcass components among commercial heavy turkey strains have been reported [3,17,141], with breast weight being especially meaningful [17,24]. However, Clayton et al. [142] found no distinction across heavy lines, and those differences were less evident compared to those of light commercial lines [17,143]. Our results might suggest a wider applicability of this trait for the comparison of heavy with light strains.

Meat protein and fat percentages were the only chemical composition traits that showed differences across breeds. The nutritional value and quality of meat depend on the species, age, carcass cut, and production system in which individuals have developed [119]. For some traits, sex also influences chemical composition [88,144]. However, the influence of genotype on the nutritional composition of meat is less clear. Attending to protein composition, some authors have reported differences across turkey breeds [17,23,119], while others did not report such differences [6,34]. In the same way, some authors have reported significant differences in meat fat content [6,17,34,119], while others reported completely different results [23]. Contradictory results are also found when comparing chemical meat composition in other poultry species [17]. It must be highlighted that studies where differences in meat protein were found compared slow-growing strains to fast-growing ones [17,23,119], while those not obtaining results only compared heavy strains [6,34]. Studies of different genotypes reared under distinct production systems have been included in this study, and hence the effect of the rearing system could have influenced meat chemical composition. In this way, physical exercise in the outdoor systems contributes to higher muscle development and enhances higher protein composition [145], while exposure to cold temperatures could consume fat meat reserves [119].

The Mahalanobis distances cladogram (Figure 4) grouped genotypes resembling a phylogenetic tree or expansion diagram of the domestic turkey. In this respect, phenomic studies have been identified that can be employed to analyze different population statuses. The Wild genotype was included in the most differentiated cluster, which is typical of primitive populations [146]. This population corresponds to the *Meleagris gallopavo sylvestris* subspecies, which has been reported to have genetically contributed to modern turkey breeds when European settlers brought the Mexican domesticated turkey to North America [147]. There is a remarkable lack of bibliography about the origin of the North Caucasian Bronze population. However, the proximity of this genotype to the Wild population described in the cladogram might suggest possible hybridization with this wild turkey population in Bulgaria, where the study was carried out [40].

Turkish Bronze and Unspecified Commercial populations are included in the next cluster, being much closer to the rest of the groups than the first one. Their possible proximity could be due to the fact that American Bronze feather heritage populations took part in the origin of modern commercial strains [147]. Those unspecified groups might comprise observations from different commercial strains, which could be specifically linked to the Bronze-feathered heritage populations. Another highlighted finding is the association between the Egyptian and Nigerian populations. This relatedness of the only two African breeds could indicate the parentage of both populations during the processes of turkey expansion through the ‘Old World’ [148,149]. 

The proximity of the Beltsville Small White to the Commercial population probably derives from the active participation of the Beltsville Small White as a heritage breed in the creation of modern commercial strains [150]. On the other hand, there is no literature explaining the origin of the Lebanese local turkey. However, its closeness to the Commercial group and Beltsville Small White suggests that this could be a commercial turkey strain from Lebanon or a breed that recently originated from these populations.

The classification tool for meat and carcass traits describes an overall efficiency of 80.62%. However, there are varied success rates among breeds. Firstly, Local Nigerian (100.00%), Wild (95.00%), Commercial (94.60%), and North Caucasian Bronze turkeys (90.00%) displayed a greater differentiation among breeds. However, tool efficiency in Beltsville Small White was 50.51%, with the remaining observations attributed to the Commercial group. This could be attributed to its very recent involvement in the formation of modern commercial strains compared to other heritage populations [150]. Moreover, 78.84% of Turkish Bronze and 88.8% of Lebanese local turkey observations were commonly classified as Commercial, while the Egyptian local turkey was 90.00% classified as Local Nigerian. These results reinforce the possible relatedness of those most commonly confused genotypes described in Figure 4. Interesting results were obtained when Unspecified Commercial observations were mostly classified as Commercial, which could suggest that this tool can be used as a genotype traceability tool.

## 5. Conclusions

The present study could be used as a guide for the evaluation of the literature resources when drawing on the experimental design of research to address the characterization of meat and carcass quality in a turkey population. Our results showed high variability in meat and carcass quality traits among worldwide turkey genotypes. However, there is a remarkable lack of studies on local breeds in comparison to commercial strains, especially from Latin America, where native breeds play a pivotal role in the rural economy and represent the origin of turkey domestication. Cold carcass weight and slaughter weight displayed the greatest discrimination power, as body weight represents the greatest phenotypic variability among turkey genotypes and explains 95% of the cold carcass weight’s variance. Nevertheless, a different response to the carcass cooling process between breeds could justify the greater discriminant ability of cold carcass weight compared to slaughter weight. Protein and fat percentages in the meat were found to statistically show discriminant power between genotypes. Mahalanobis distances analysis showed interesting clustering patterns, grouping potential phylogenetically related populations. Therefore, this DCA could be employed as a genotype traceability tool and an additional tool for guidance on tracing population relatedness in case of scarce resources. Additionally, this study could be the starting point for the development of a genotype classification tool based on phenomic traits.

## Figures and Tables

**Figure 1 foods-12-03828-f001:**
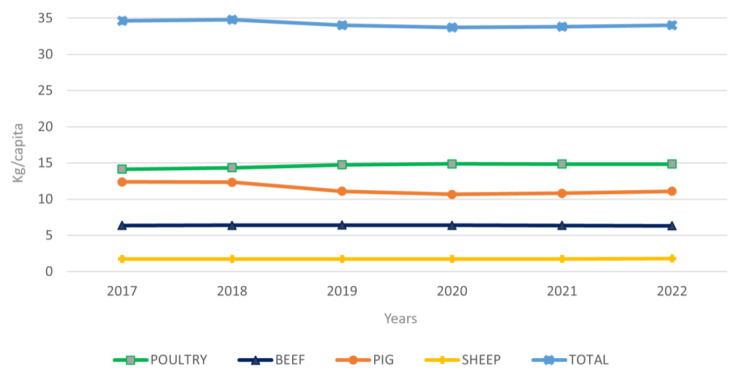
Evolution of meat consumption and linear tendencies (kg/capita). Information source: OCED [4].

**Figure 2 foods-12-03828-f002:**
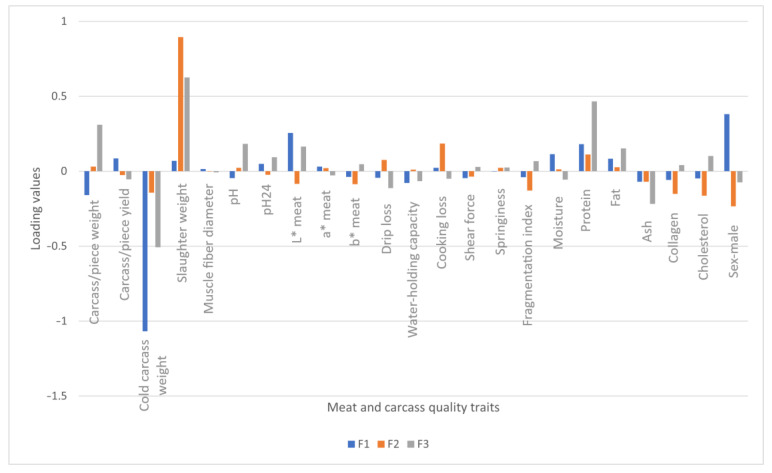
Loading values of each meat and carcass quality traits in functions F1, F2, and F3.

**Figure 3 foods-12-03828-f003:**
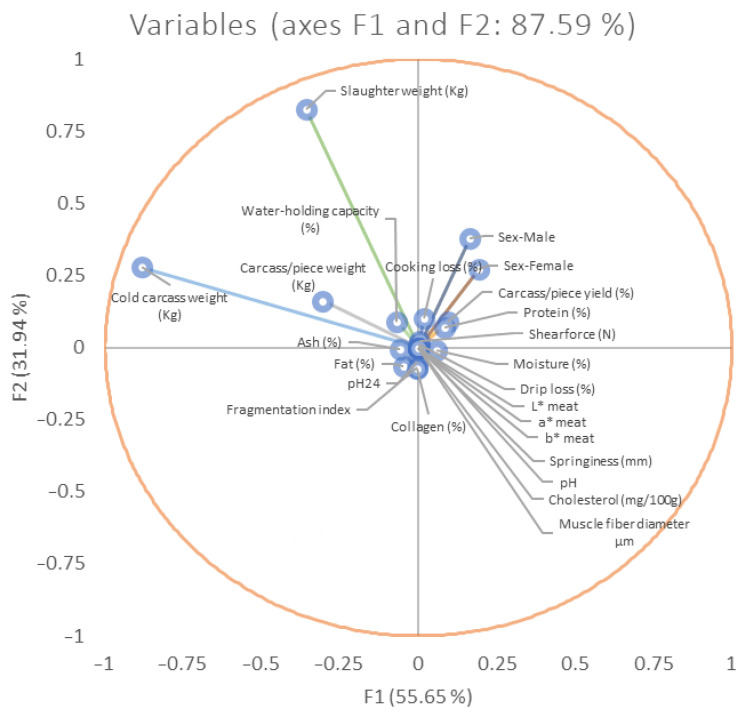
Vector plot for discriminant loadings for meat- and carcass quality-related traits.

**Figure 4 foods-12-03828-f004:**
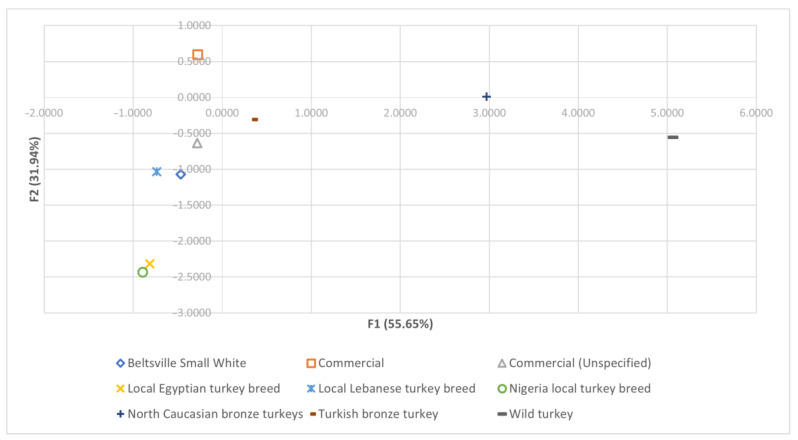
Territorial map depicting the centroids considered in the DCA sorted across the studied turkey breeds.

**Figure 5 foods-12-03828-f005:**
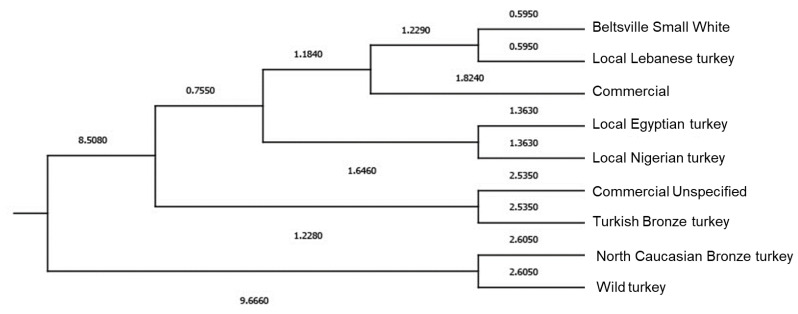
Tree diagram based on Mahalanobis distances between carcass and meat quality traits and turkey breeds.

**Figure 6 foods-12-03828-f006:**
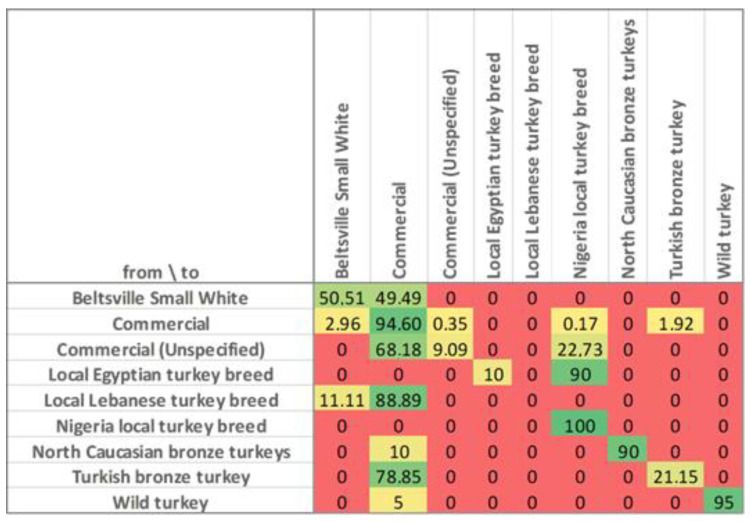
Graphical depiction of the confusion matrix for the percentages (%) of the correct assignment of the observations in each group obtained via the classification and leave-one-out cross-validation matrices. This heat map uses a range of warm and cool colors, indicating respectively lower or higher percentage of correct assignment. Warmer colors (red and yellow tones) indicate a lower percentage of correct assignment and cooler colors (green tones) indicate a higher percentage of correct assignment.

**Table 1 foods-12-03828-t001:** Clusters, references, and units of the traits analyzed in this study.

Cluster	References	Trait	Unit
Carcass dressing traits	[6,10,17,32,33,34,35,36,37,38,39,40,41,42,43,44,45,46,47,48,49,50,51,52,53,54,55,56,57,58,59,60,61,62,63,64,65,66,67,68,69,70,71,72,73,74,75,76,77,78,79,80,81,82,83,84,85,86,87]	Carcass/piece weight Carcass/piece yield Cold canal weight Slaughter weight	kg % kg kg
Muscle fiber properties	[17]	Muscle fiber diameter	μm
pH	[6,10,16,17,35,43,44,54,55,56,57,59,67,68,69,70,76,77,79,80,81,82,83,85,86,88,89,90,91,92,93,94,95,96,97,98,99,100]	pH pH 24 h	
Color-related traits	[6,10,17,34,35,37,43,44,54,55,57,59,67,68,70,76,77,79,83,85,86,88,89,91,93,94,98,99,100,101,102]	L* meat a* meat b* meat	
Water-retaining characteristics	[6,16,17,35,37,43,44,45,47,53,54,55,56,57,59,68,70,76,79,80,82,83,85,86,88,89,90,92,93,94,95,96,97,98,100,101,102]	Water holding capacity Drip loss Cooking loss	% % %
Texture-related traits	[6,16,17,43,53,54,57,68,76,79,80,83,85,89,90,93,95,97,98]	Shear force Springiness Fragmentation index	N cm gr
Meat chemical composition	[6,17,34,35,42,43,51,53,55,56,57,58,59,62,66,68,76,81,82,83,85,88,89,90,92,94,95,97,98]	Moisture Protein Fat Ash Collagen Cholesterol	% % % % % mg/100 g

**Table 2 foods-12-03828-t002:** Summary of the results of Pillai’s trace of equality of covariance matrices of canonical discriminant functions, used to determine the suitability of data for performing DCAs’ discriminant canonical analyses.

Parameter	Value
Pillai’s trace criterion	1.3676
F (Observed value)	7.4058
F (Critical value)	1.1766
df1	192
df2	6896
Significance	<0.0001
Alpha	0.05

**Table 3 foods-12-03828-t003:** Collinearity checking analysis for meat- and carcass quality-related traits.

Statistic	Tolerance (1 – R^2^)	VIF ^1^
Carcass/piece weight (kg)	0.7378	1.3554
Carcass/piece yield (%)	0.7967	1.2551
Cold carcass weight (kg)	0.6079	1.6450
Slaughter weight (kg)	0.5766	1.7342
Muscle fiber diameter (µm)	0.9943	1.0058
pH	0.5052	1.9796
pH24	0.6467	1.5464
L* meat	0.7291	1.3716
a* meat	0.7453	1.3418
b* meat	0.6819	1.4664
Drip loss (%)	0.6159	1.6236
Water-holding capacity (%)	0.6938	1.4413
Cooking loss (%)	0.8127	1.2305
Shear force (N)	0.6766	1.4779
Springiness (mm)	0.9927	1.0074
Fragmentation index	0.7489	1.3353
Moisture (%)	0.5812	1.7207
Protein (%)	0.6502	1.5380
Fat (%)	0.4011	2.4934
Ash (%)	0.4653	2.1490
Collagen (%)	0.4599	2.1742
Cholesterol (mg/100 g)	0.3797	2.6336
Sex-male	0.6345	1.5760
Sex-female	0.6959	1.4369

^1^ Interpretation thumb rule: variance inflation factor (VIF) = 1 (not correlated); 1 < VIF < 5 (moderately correlated); VIF ≥ 5 (highly correlated).

**Table 4 foods-12-03828-t004:** Canonical variable functions and percentages of cumulative variance.

	F1	F2	F3	F4	F5	F6	F7	F8
Eigenvalue	1.522	0.873	0.201	0.085	0.025	0.020	0.006	0.002
Cumulative variability (%)	55.649	87.594	94.930	98.040	98.955	99.693	99.919	100

**Table 5 foods-12-03828-t005:** Results of the tests of the equality of groups’ means.

Variable	Rank	Lambda	F	DF1	DF2	*p*-Value
Cold carcass weight (kg)	1	0.487974954	115.1590844	8	878	<0.0001
Slaughter weight (kg)	2	0.590140576	76.22263836	8	878	<0.0001
Sex-male	3	0.905514267	11.45184512	8	878	<0.0001
Carcass/piece weight (kg)	4	0.906325972	11.34329683	8	878	<0.0001
Protein (%)	5	0.937494654	7.317334257	8	878	<0.0001
Fat (%)	6	0.943780019	6.537691868	8	878	<0.0001

## Data Availability

The data used to support the findings of this study can be made available by the corresponding author upon request.

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
