# Peer review of "Study of Meat and Carcass Quality-Related Traits in Turkey Populations through Discriminant Canonical Analysis"

_foods, 2023, doi:10.3390/foods12203828_

Round 1

Reviewer 1 Report

The authors present a meta-analysis of the data gathered from 75 articles to identify discriminant features among 9 turkey breeds. They used Canonical Discriminant Analysis as the main tool to achieve their goal.

My overall impression is positive. I appreciate the "data reuse" approach presented by the authors, it is clearly a relevant data science strategy. In other context, it can also be seen as an alternative to animal slaughtering for research purpose: reuse and reanalyze existing data instead of generating new ones. This is not an easy task and the authors probably did a big work to "make observations accross papers comparable".

But, I still have a some comments to improve the readability of the paper mainly concerning Mat & Meth.

# Major comments

- Dataset: the number of variables is rather clearly set:
  * 22 numerical variables meat and carcass + age at slaughter
  * 2 categorical (variety, sex)
  * 1 categorical outcome (breed)
but it's difficult to know how many observations are in the dataset. Between 50 and 2500 from section 2.3.1, but never set explicitely or is it 75 as the number of documents? The authors should clarify this point.

- The workflow used to analyze the data is not clear for me. Several methods are described in the Mat & Meth sections and I cannot figure out how they complete each other: Bayesian ANOVA, multicollinearity, CDA, multinomial logistic regression for variable selection, PCA for dimensionality reduction. Specifically, several methods aim to select variables (multicollinearity checking, multinomial logistic, PCA) whereas the section Results does not seem to reveal any variable selection.

- I would also appreciate that the authors justify better the use of not so standard method (Bayesian ANOVA and Discriminant Canonical Analysis) and their complementarity.
  * the authors should justify the use of Bayesian ANOVA instead of Kruskall-Wallis for instance. Standard ANOVA can indeed be ignored because of non normality. And a reference should be given because Bayesian ANOVA is not a so standard method. Could the authors confirm that Bayesian ANOVA detects potential differences *in the median*? It is not obvious according to what I've read.
  * The sentence "Hence, the presence of differences in some variables accross breeds justified the employment of a Discriminant Canonical Analysis". In my mind, a multivariate discriminant method such as DCA could also be justified when a univariate approach such as (Bayesian) ANOVA failed to detect differences. Can the authors discuss this point? The authors could also add a reference about DCA.

- Multicollinearity is not necessarily a problem. Depending on the goal of the analysis, it can be resolved or not. The authors should justify they use it to "exclude unnecessary variables" (line 175), even if actually, no variable seems to suffer from multicollinearity (Results, table 3).

- Section 2.3.4 DCA. I don't understand why the authors refer to variable selection using "Regularized forward stepwise multinomial logistic regression algorithms". The discriminant functions provided from line 265 seem to consider every dependent variables previously mentionned. And what would be the interest of using both DCA and multinomial logistic regression, two methods that can be considered to have the same goal.

- Section 2.3.6. I don't understand the use of PCA for dimensionality reduction as, once again, all the variables seem to be retained in the Discriminant Canonical Analysis.

# Minor comments

- Figure 1 is not so easy to read and should be reformated:
  * maybe two vertical axes are not mandatory as the order of magnitude are nearly the same
  * barplot mixed with lines does not make easier the readibility
  * maybe only lines would be enough, possibly with dots in different symbols and colors, as for instance what can be seen here: https://statisticsglobe.com/plot-line-in-r-graph-chart#seventh

- line 155: section 2.3.1 is intitled "Normality and Kruskall-Wallis tests" but it seems that no Kruskall-Wallis has been performed. The authors should clarify this point or change the title.

- From line 265: the definition of the first three discriminanting functions is difficult to read. Maybe a table or a graphical display of the loading values could be considered. See for instance loading plots in PCAtools and mixOmics packages both available on bioconductor.org.

- Figure 3: could the authors provide a graphical output of the individuals? I would be nice to go with the interpretation of the vector plot in Figure 3.

- Line 302. I don't understand the sentence "Mahalanobis distances obtained... squared Euclidean distance". Could the authors clarify this point?

- Figure 5: the figure is difficult to read, in particular, colors are not so easy to distinguish. Maybe the figure could be replaced by a confusion matrix (possibly represented as an image if the authors prefer to keep a figure).

# Typos or very minor comments

- line 103: the sentence "The approach ... has been approached..." could be reworded to avoid using "approach" twice.

- line 111 and elsewhere: homogeneize citations without the name of the authors

- line 186: 27 parameters? shouldn't it be 22 (line 132)?

- line 215: notation >= |0.40| is a bit weird. Probably, the meaning is "the absolute value of a discriminant loading is higher than 0.40"

- line 252: the paragraph "The preliminary ... in Table 3" seems to be written as a legend (small size font).

- line 255: is the legend of Table 3 consistent with its content? It should mention something like "collinearity checking".

- Maybe Figure 2 could be removed as it is exactly redundant with the table provided at the bottom of the figure. The table only would be enough.

Reviewer 2 Report

The study is focused on the meat and carcass quality related traits in turkey genotypes worldwide and aims to find the differences in these traits applying meta-analytical approach. In the Introduction, the authors provide sufficient information, also schematized to justify their aim and to convince this study is important to the readers (both scientists and professionals).

The methodology is described in sufficient details. However, in the data collections the authors state that "On the other hand, other information sources as www.ncbi.nlm.gov/pubmed/ were discarded since they do not include tools enabling data extraction for analysis, contrary to those previously mentioned" It is not clear which tools the authors are referring to. Are these statistical parameters such as standard deviation, standard errors??? Please, explain.

The time span of the documents is also very large (1968-2021). Why have the authors considered including such old documents in the meta-analysis?

In line 153 the authors state: "In addition, breed, variety, sex, age at slaughter, and production system were taken into account at the time of data collection". When referring to breed, this is clear, because it was said above that the breeds included in the analysis are 9. However, for the other factors it remains unclear how exactly they were taken into account? Were the turkeys at the same age, what are the specifics of nutrition as well, since it is known that diet influences the meat quality.

Apart from these remarks, the results of the meta-analysis and the accompanying test are very clearly described and discussed. The number of tables and figures is adequate. The conclusions are rather detailed. Might be more concise, but I will leave this to the authors.

Round 2

Reviewer 1 Report

The authors have taken most of my comments into account but I still have minor comments.

- I do not fully agree with this sentence "Linear discriminant function analyses perform a multivariate test of differences between groups". Presenting LDA as a way to perform a multivariate test could be misleading for the reader. I would not say that one has to use LDA to perform multivariate test.

- the sentence "Therefore, the maximum likelihoods of the likelihood distributions..." is not clear for me. Maybe it's because of the use of the word "likelihood" three times with potentially different meaning (I'm not sure...)

- the justification of using PCA remains unclear for me. "data are first transformed using a PCA and subsequently, clusters are identified using a DCA". To which extent, PCA tranforms the data, and how DCA allows to identify clusters as it is a supervised method with pre-defined groups.

- Figure 2 does not convince me: colors are difficult to distinguish and the figure is strictly unreadable for colorblind people. If the authors does not find another way to represent the loadings (for instance something like 3 barplots - one for each function Fi - with as many bars as variables, the height of the bar representing the loading), I suggest they switch back to the equation of version 1, maybe using a True Type font to enhance the readability.

- Figure 4 present the territorial map of the centroids. I would have liked to see all the individuals on a plot, but I can imagine the plot would not be so easy to read, so I'm OK.
